# JUUL™ing and Heating Lead to a Worsening of Arterial Stiffness

**DOI:** 10.3390/medicines9040028

**Published:** 2022-04-05

**Authors:** Julia Benthien, Moritz Meusel, Silja Cayo Talavera, Ingo Eitel, Daniel Drömann, Klaas F. Franzen

**Affiliations:** 1Medical Clinic III, Campus Lübeck, University Hospital Schleswig-Holstein, 23562 Lübeck, Germany; julia.benthien@student.uni-luebeck.de (J.B.); silja.cayotalavera@uksh.de (S.C.T.); daniel.droemann@uksh.de (D.D.); 2Airway Research Center North, Member of the German Center for Lung Research (DZL), 22927 Großhansdorf, Germany; 3Medical Clinic II, Campus Lübeck, University Hospital Schleswig-Holstein, 23562 Lübeck, Germany; moritz.meusel@uksh.de (M.M.); ingo.eitel@uksh.de (I.E.)

**Keywords:** arterial stiffness, endothelial dysfunction, JUUL, HTP, cigarette, e-cigarette, smoking, vaping, heating, cardiovascular risks

## Abstract

**Background**: The widespread use of the JUUL™ device ignited a discussion about the effects these products have on harm reduction. Therefore, we conducted a study directly comparing the JUUL™ device with a cigarette, a heated tobacco product, and a nicotine-free e-cigarette to examine the acute effects on arterial stiffness. **Methods:** This crossover-designed study examines 20 occasional smokers (age 25.2 ± 2.5 years). Study participants used each of the four smoking devices for a duration of 5 min following a protocol. Peripheral blood pressure and parameters of arterial stiffness and endothelial vasodilator function such as the reactive hyperemia index and the augmentation index were measured using the EndoPAT™2000 before and after. **Results:** In addition to significant peripheral hemodynamic changes after 5 and 10 min (*p* < 0.05), the reactive hyperemia index showed a significant decrease for all devices 15 min after consumption and remained significantly decreased after 60 min (*p* < 0.01). The augmentation index adjusted for a heart rate of 75 bpm increased significantly for all devices 15 and 60 min after consumption (*p* < 0.01). **Conclusions:** In conclusion, the increases in blood pressure and arterial stiffness are similar after smoking, JUUL™ing, heating, and vaping. These changes may be associated with an increase in cardiovascular risks; however, an evaluation of the long-term effects of JUUL™ing, vaping and heating is needed.

## 1. Introduction

“JUUL™ing”, as the use of the JUUL™ device is called by young people, was the most popular way of using an e-cigarette among youth in the U.S. in 2017, only two years after the launch of the JUUL™ device on the U.S. market [1]. At the end of 2018, the JUUL™ device became the best-selling e-cigarette in the U.S., accounting for about 75% of the e-cigarette retail market share there [2]. During the same year, an increase of 78% of the current e-cigarette users among American high-schoolers was recorded, reaching a ratio of more than 20% of all students [3]. Although companies marketed the products as a smoking cessation tool, e-cigarettes are only rarely used for this purpose [4]. Despite the widespread use of the JUUL™ device among youth, adolescents are often uninformed about the constituents of the inhaled substance: 63% of 15- to 24-year-olds surveyed were unaware that JUUL™ products always contain nicotine [5]. This is alarming, especially considering that a JUUL™-pod, as is called the cartridge that contains the liquid, is reported to contain substantially more nicotine than the majority of e-cigarettes on the market with an amount of up to 59 mg/mL of nicotine per pod sold in the U.S. [6], whereas the maximum concentration of nicotine in e-liquids allowed in the European Union is 20 mg/mL [7].

Considering the harmful effects of nicotine on developing brains in combination with the relatively high concentration of nicotine used in the JUUL™ device that, due to the nicotine salt formulation [8,9], increases even higher once delivered to the bloodstream, there are many concerns about the use of the JUUL™ device among young people. The data show that younger users are more likely to become addicted, have more difficulty quitting, and might be at higher risk for addiction to other substances in the future [10,11]. This phenomenon is called the “gateway” hypothesis of e-cigarettes, suggesting that vaping might be an entry point to combustible cigarettes and cannabis [4].

Despite the fact that there still is a lack of information about the health risks that can be associated with the use of e-cigarettes [12], in October 2019, a severe consequence of the use of e-cigarettes emerged: “EVALI”, an acronym for “e-cigarette or vaping product use-associated lung injury” [13]. “EVALI” is used as an umbrella term to describe lung diseases associated with the use of vaping products that meet the following criteria: use of e-cigarette devices and related products in the 90 days before symptom onset and pulmonary infiltrates on imaging and absence of pulmonary infection on initial work-up and no evidence in medical record of alternative plausible diagnoses (e.g., cardiac, rheumatologic, or neoplastic process) [14]. Laboratory data have shown that the majority of the EVALI cases can be associated with THC-containing e-cigarettes as well as vitamin E acetate, which is used as an additive in some THC-containing vaping products [15]. However, longitudinal data assessing the association between e-cigarette use of young adults and respiratory symptoms revealed that e-cigarette use, independent of the THC content, was associated with higher odds of developing wheezing-related respiratory symptoms, after accounting for cigarette smoking and other combustible tobacco product use [16]. In response to severe cases of lung illness with an acute respiratory distress syndrome, the FDA passed a legislative decision in January 2020 that tobacco products may only be sold to adults aged 21 years and older [17].

The increased use of e-cigarettes among adolescents in the U.S. has led to new interest regarding the long-term harm of e-cigarettes [18]. Smoking is still one of the leading causes of mortality in Germany and other western countries [19].

Most e-cigarettes produce significantly lower emissions of hazardous and potentially hazardous substances compared to cigarette smoke [20,21]. The use of alternative smoking devices, such as heated tobacco products (HTP) and especially e-cigarettes, are expected to reduce the risks for some diseases, particularly lung cancer. For this reason, e-cigarettes, including the JUUL™ device, were suggested to be used as tools for smoking cessation [22]. However, to evaluate the reliability of this recommendation, further research is needed. The assumption that e-cigarettes reduce the rate of cardiovascular disease must be challenged, since they also deliver nicotine. In several studies, the chronic effects of smoking on cardiovascular events have been evaluated. Smoking represents one of the most important cardiovascular risk factors [23,24] and at the same time one of the most modifiable ones [25].

To assess subclinical end-organ damage, the European Society of Hypertensiology and the European Society of Cardiology recommend measuring arterial stiffness and endothelial dysfunction, which are parameters that also can indicate the onset of systemic cardiovascular diseases [26]. Since endothelial dysfunction is an early predictor of cardiovascular disease, the EndoPAT™2000 system (Itamar Medical Ltd., Caesarea, Israel) is a suitable and verified device that measures the parameters of arterial stiffness and endothelial vasodilator function [27]. Several studies have examined e-cigarettes and HTP concerning the effects on arterial stiffness and showed evidence that e-cigarettes can increase cardiovascular risks [28,29].

Thus, frequent, and particularly daily e-cigarette use increases the risk of myocardial infarction, as published by the working group of Alzahrani [30]. Contrary to these findings, a recent epidemiological analysis did not confirm increased cardiovascular risks for vapers who never smoked cigarettes [31]. Especially for devices that deliver high levels of nicotine, such as the JUUL™ [32] or heated tobacco products [33], evidence is still limited [34]. More data about the effects of JUUL™ing on the pathogenesis of cardiovascular diseases, specifically on arterial stiffness and endothelial dysfunction, is required. In their model about the pathogenesis of endothelial dysfunction triggered by e-cigarette vapor, Lee et al. postulate the inflammatory response as a cause of endothelial dysfunction in addition to apoptosis and reactive oxygen species [35].

The purpose of this study was to investigate the acute effects of the JUUL^TM^ device on peripheral blood pressure, arterial stiffness, endothelial vasodilator function, and inflammatory response. To rank the measured effects against other commonly used devices such as the combustible cigarette or the HTP, we performed a direct comparison of these three devices including a control group without nicotine.

## 2. Materials and Methods

### 2.1. Study and Cohort Design

The study was designed as a single-center, randomized, four-arm study with a crossover design including 20 young, occasional smokers, or e-cigarette users with both sexes represented equally. Participants were recruited from the campus of the University of Lübeck and were screened based on the following inclusion criteria: (I) smoker or vaper, (II) no mental disorder, (III) no cardiovascular disease, (IV) no thyroid disease, (V) no diabetes, (VI) no abnormalities in physical examination, (VII) no hypertension, and (VIII) no hypercholesterolemia. To exclude possible pregnancies, women were only accepted as participants if they were applying a reliable contraceptive method. In addition to this criterion, participants were excluded if they declared that they were strict non-smokers or that they participated in any other kind of study at the same time.

During the screening, all potential participants received a document of informed consent for the study, which needed to be signed after a consideration time of at least 24 h to become a study participant. The participants were asked to follow the guidelines for measuring arterial stiffness [36,37]. To do so, the consumption of nicotine by smoking cigarettes as well as e-cigarettes or other devices and drinking alcohol was prohibited for 24 h before the test appointment. The time interval of not smoking was controlled by measuring the CO-content of the exhaled air with a Micro+Smokerlyzer™ (Bedfont Scientific Ltd., Maidstone, UK) with a cut-off of 6 ppm carbon monoxide. At the end of the screening appointment, the order of the four different test conditions was randomized for each participant by drawing lots. The study was approved by the ethics committee of the University of Lübeck by an amendment to a prior study on the topic of e-cigarettes (AZ1717).

The four different study arms consisted of the following products: (a) a commercial, combustible tobacco cigarette (Cig group; Marlboro Gold, Philip & Morris, 0.5 mg nicotine), (b) a JUUL™ e-cigarette with first-generation technology equipped with a JUUL™-pod of the flavor “Virginia Tobacco” (JUUL group; 20 mg/mL nicotine), (c) an IQOS™ as a heated tobacco product equipped with a HEET™ of the flavor “BRONZE” (HTP group; 0.5 mg nicotine per HEET), and (d) an e-cigarette without nicotine (device: DIPSE e-cigarette, Original eGo-T CE4 vaporizer, third generation, 3.3 volts, 1.5 ohms, 7.26 watt; liquid: DIPSE, Tobacco Premium Line Liquid, nicotine-free) (E-Cig (-) group; 0 mg nicotine). An e-cigarette is a handheld battery-powered device that generates an aerosolized vapor from a liquid typically consisting of nicotine, propylene glycol (PG), vegetable glycerin (VG), and flavoring chemicals [38]. The JUUL™ device is an e-cigarette with a compact design that resembles a USB flash drive consisting of a one-time use cartridge, which includes the mouthpiece as well as the heating element, called a “JUUL^TM^-pod”. This is put on the corpus of the e-cigarette that stores the chargeable battery. Specific for the JUUL™ device is the use of nicotine salt-based liquid. The IQOS^TM^, in the following referred to as the heated tobacco product (HTP), contains tobacco leaf in form of a heat stick, which is heated at a lower temperature than conventional cigarettes [39], producing aerosols containing nicotine and toxic chemicals as well as non-tobacco additives, often including flavors [40].

All four test conditions needed to be fulfilled by each participant to complete the study. In total, 83 single measurements were performed, but only the data of these 20 participants who fulfilled all four test conditions were included in the statistical analysis. Three measurements were considered dropouts because the respective participants were excluded throughout the study due to individual reasons. Therefore, the data of 80 measurements are now represented in the results of this study.

Each test day was performed in the following sequence: In the beginning, the CO content of the exhaled air was measured to check the no-smoking rule before the test appointment. Before the measurements started, the participants were introduced to the use of the respective device and were trained to vape the two types of e-cigarettes along a schema of 1 puff every 30 s for 5 min, with a duration of 3 s per puff, as described in other publications [41]. The combustible cigarette and the heated tobacco product were consumed normally until completed.

Generally, measurements were started at least 30 min before smoking, JUUL™ing, or vaping. At first, a peripheral venous catheter was placed in one of the arm veins to take blood samples at several points of time during the test day. A blood sample to obtain a hemogram was taken directly at the beginning and two hours after smoking, JUUL™ing, or vaping. Further blood samples were taken for cortisol determination by high-pressure liquid chromatography at the following time points: in the beginning, after 10 min, and two hours after using the respective device, based on the design of a previous study showing significant changes of the cortisol concentration in plasma after smoking cigarettes [42]. The peripheral blood pressures and heart rate were measured every 5 min from 30 min before and until 75 min after smoking, JUUL™ing, or vaping by the experimenter using a conventional electric oscillometric blood pressure monitor (Omron MIT Elite Plus™, Omron, Kyoto, Japan). The endothelial vasodilator function expressed as the reactive hyperemia ratio (RHI), as well as the arterial stiffness, represented by the augmentation index (AI), were analyzed utilizing the EndoPAT™2000 system (Itamar Medical Ltd., Caesarea, Israel). These parameters were measured at three points of time during each test day: once 15 min before and two times 15 min and 60 min after device use. All four measurements of one study participant were performed at the same time of day to avoid biases due to circadian rhythm.

### 2.2. Measurement of Endothelial Vasodilator Function and Arterial Stiffness

Endothelial vasodilator function and arterial stiffness were measured by an EndoPAT™2000 (Itamar, Israel). The EndoPAT™ system detects via plethysmography pressure changes in the fingertips caused by the arterial pulse wave and translates this to a peripheral arterial tone. Peripheral artery tonometry (PAT) detects changes in fingertip volume as a measurement of pulse wave amplitude (PWA) over time.

Endothelium-mediated changes in vascular tone after occlusion of the brachial artery reflect a downstream hyperemic response, which is a measure for arterial endothelial function [43]. Measurements on the contralateral arm are used to control for concurrent nonendothelial-dependent changes in vascular tone [27]. The augmentation index is adjusted for a heart rate of 75 beats per minute (bpm). After participants had rested for at least 10 min, measurements were started a minimum of 15 min before they began administering the respective device in a sitting position. These data were used as a baseline and references for statistical analyses.

### 2.3. Power Calculation and Statistical Analyses

The calculation of the power is carried out via the program GPower (version 3.1). The analysis revealed four subjects, based on the results for the augmentation index from the work of the working group led by Adamopoulos (4.2, −0.7, and an average standard deviation of 1.3, with an alpha of 0.05 and a power of 0.8) [44]. Based on the experience of our own studies and the study regarding nicotine tablet vs. passive smoking, the decision was made within the working group to increase the n-number in a four-arm experimental setup to a total of 20 subjects [28,33].

Statistical analyses and graph editing were performed with SPSS statistical software (SPSS 23 Inc., Chicago, IL, USA) and GraphPad Prism 4 (GraphPad Software Inc., San Diego, CA, USA), respectively. Baseline mean values of blood pressure and arterial stiffness were used as reference values. Before further analysis, all data, e.g., peripheral and central blood pressures, were analyzed for normal distribution by Kolmogorov–Smirnov tests. Where statistically indicated, we analyzed the data with a paired Student *t*-test or Wilcoxon rank-sum test of baseline measurement vs. Nth measurement. Due to the longitudinal design, we used a two-way repeated-measures analysis of variance (ANOVA) to evaluate the effect of time and device. In addition to these two-way repeated-measures analyses of variance, we did not perform multiple testing corrections for different outcomes. Accordingly, if we found no interaction, we performed post hoc tests (Bonferroni) with G*Power. Where applicable, we performed a multivariate analysis of variance (MANOVA) correcting for age, SBP, heart rate (HR), and sex. If not otherwise stated, all data are expressed as mean (SED). A *p*-value < 0.05 was considered statistically significant.

## 3. Results

### 3.1. Study Design and Baseline Characteristics

The flowchart of the study design is shown in Figure 1. Baseline characteristics for all analyzed 20 participants are presented in Table 1.

### 3.2. Cigarette, HTP, and E-Cigarette Increased CO in Exhalation

The CO was normally distributed, and therefore, a paired Student *t*-test was used to analyze the significance between the first (baseline) and Nth measurement. The CO in exhaled air was elevated for the combustible cigarette (*p* < 0.01), HTP (*p* < 0.05), and e-cigarette (*p* < 0.05) experimental conditions until 60 min after use and remained significantly elevated for another 60 min for the cigarette (*p* < 0.01) and e-cigarette (*p* < 0.01) (Figure 2). For HTP, there was an increase as a trend without reaching significance (*p* = 0.07, Figure 2). For JUUL™, there was no significant increase across follow-up (*p* > 0.05). Analysis of variance for repeated measurements showed a significance over time (*p* < 0.05) but not device (*p* > 0.05).

### 3.3. Peripheral Systolic Blood Pressure and Diastolic Blood Pressure

The peripheral systolic blood pressure (pSBP; normal distributed) increased significantly by more than 7% directly after completed device use of the cigarette, JUUL™, and HTP (*p* < 0.05; Figure 3). At this point of time, the four groups differed significantly (*p* < 0.05). After 5 and 10 min, all four groups were significant elevated (*p* < 0.05; Figure 3). During the follow-up, pSBP was significantly elevated within the HTP group at several time points (Figure 3). Analysis of variance for repeated measurements of pSBP showed a significance over time (*p* < 0.05) and device (*p* < 0.05).

At the initial, 5-min, 10-min, and 15-min measurement, peripheral diastolic blood pressure (pDBP; normal distributed) increased by at least 6% in the groups of cigarette (*p* < 0.01), JUUL™ (*p* < 0.01), and HTP (*p* < 0.01) (Figure 4). The pDBP was significantly increased in the e-cigarette group after 5 and 10 min (*p* < 0.05; Figure 4). The four groups differed significantly from each other for the first five minutes (ANOVA; *p* < 0.01). Analysis of variance for repeated measurements of pDBP showed a significance over time (*p* < 0.05) and device (*p* < 0.05). Peripheral pulse pressure (pPP; normal distributed) did not change significantly during the follow-up in any group (*p* > 0.05; Figure 5). All four groups did not differ from each other at any follow-up time-point (*p* > 0.05; Figure 5). Analysis of variance for repeated measurements of pPP did not show a significance over time (*p* > 0.05) and device (*p* > 0.05).

### 3.4. Increase in the Heart Rate in the Groups of Cigarette, JUUL™, and HTP

For the first 5 min, the heart rate (HR; normal distributed) increased significantly by more than 9% in the groups of the cigarette, JUUL™, and HTP (each *p* < 0.05; Figure 6), and the heart rate remained significantly increased for 5 more minutes in the group of cigarette and HTP (each *p* < 0.05; Figure 6). The e-cigarette group did not increase at any point in time (*p* > 0.05). In the groups of JUUL™ and the e-cigarette, there were isolated significant drops in heart rate during further follow-up periods (Figure 6). The four groups differed significantly for the first 10 min in the ANOVA (*p* < 0.01). Analysis of variance for repeated measurements of HR showed a significance over time (*p* < 0.05) and device (*p* < 0.05).

### 3.5. Aggravation of the Augmentation Index and Reactive Hyperemia Index after Using Any Type of Device

The reactive hyperemia index (RHI; normal distributed) showed a significant change after 15 min in the groups of cigarette (−17%; *p* < 0.05), JUUL™ (−22%; *p* < 0.01), HTP (−16%; *p* < 0.01), and e-cigarette (−21%; *p* < 0.01) (Figure 7; Table 2). After 60 min, the decreases of RHI remained significant in all four groups (*p* < 0.01; Figure 7, Table 2). The four groups differed after 15 min significantly within the ANOVA (*p* < 0.05; Table 2). Analysis of variance for repeated measurements of RHI showed a significance over time (*p* < 0.05) and device (*p* < 0.05)

After performing a multivariate analysis of RHI including blood pressure and heart rate, all three groups’ changes remained significant (Box test, each *p* > 0.05; Levene test, each *p* < 0.05; Wilks’ lambda (multivariate test), each *p* < 0.05). The values for the natural logarithm of the RHI (lnRHI) showed a comparable course and significance to the RHI (Figure 8, Table 2).

The augmentation index adjusted for a heart rate of 75 bpm (AI@75bpm; normal distributed) increased significantly in the groups of cigarette (+32%; *p* < 0.01), JUUL™ (+33%; *p* < 0.01), HTP (+32%; *p* < 0.01), and e-cigarette (+26%; *p* < 0.05) after 15 min (Figure 9, Table 2). AI@75bpm remained more than 30% higher after 60 min and reached significance (*p* < 0.01; Figure 9, Table 2). There were no significant differences between the four groups at any point of time. Analysis of variance for repeated measurements of AI@75 showed a significance over time (*p* < 0.05) and device (*p* < 0.05). After performing a multivariate analysis of AI@75bpm including blood pressure and heart rate changes, the results remained significant in all four groups (Box test, each *p* < 0.05; Levene test, each *p* < 0.05; Wilks’ lambda (multivariate test), each *p* < 0.05).

### 3.6. Decrease in Plasma Cortisol Concentration and Changes within the Hemogram after Using the Devices

The plasma cortisol concentration (normal distribution) decreased significantly after 12 min in the JUUL™ group (−15%; *p* < 0.05, Figure 10; Table 3). After 120 min, the decrease in cortisol became highly significant in all four groups (*p* < 0.01; Figure 10, Table 3). The four groups did not differ at any point of time in the ANOVA during the follow-up (*p* < 0.05; Table 3). Analysis of variance for repeated measurements of cortisol showed a significance over time (*p* < 0.05) and device (*p* < 0.05).

Leucocytes, which were not normally distributed (Wilcoxon test), significantly increased 120 min after the use of each device in the groups of JUUL™ (*p* < 0.05), HTP (*p* < 0.01), and e-cigarette (*p* < 0.05) (Figure 11, Table 4).

For immature granulocytes, which were not normally distributed (Wilcoxon test), there was a significant increase in the JUUL™ group (*p* < 0.05; Figure 12, Table 4) and a trend without reaching the significance level in the cigarette group (*p* = 0.058; Figure 12, Table 4).

Regarding the neutrophil granulocytes, which were also not normally distributed (Wilcoxon test), there was a significant increase in the HTP group (*p* < 0.01; Figure 13, Table 4) and the e-cigarette group (*p* < 0.05; Figure 13, Table 4) as well as trends without reaching significance for the cigarette group (*p* = 0.051; Figure 13, Table 4) and the JUUL™ group (*p* = 0.057; Figure 13, Table 4).

## 4. Discussion

Although the JUUL™ is a widely used device, the knowledge about the effects of JUUL™ing on arterial stiffness is still limited. This is accompanied by a lack of awareness among the population of JUUL™ing as a potential risk factor for cardiovascular disease and as a device that can lead to addiction due to its nicotine content [45]. Considering the current state of research [46,47,48,49,50], this is the first direct comparison of the impact of JUUL™ing, using a heated tobacco product, nicotine-free e-cigarette use, and cigarette smoking on peripheral hemodynamics, endothelial vasodilator function, and arterial stiffness.

Our trial demonstrates acute changes of the peripheral blood pressure as well as changes in the parameters of endothelial vasodilator function and arterial stiffness for a short period after JUUL™ing, tobacco heating, vaping, and smoking. Studies published so far lead to contradictory statements about the acute effects of vaping, tobacco heating, and smoking on arterial stiffness. Some have been able to demonstrate an increase in arterial stiffness and thus arterial deterioration, and others have revealed no change [28,33,47,48,51]. In line with our previous studies on e-cigarettes, the group of Vlachopoulos showed a negative impact of vaping on central hemodynamic and peripheral blood pressure [28,48]. As expected, and already published [28,47,48,51], heated tobacco products, cigarettes, and e-cigarettes with nicotine-containing liquids lead to an increase in peripheral and central hemodynamic parameters and arterial stiffness. Nevertheless, the acute effects of JUUL™ on arterial stiffness were still unknown until now.

Regarding the JUUL™ device, we recognized a significant increase in the peripheral blood pressure directly after consumption, remaining significantly increased for 10 more minutes, comparable to the changes found after consumption of the heated tobacco product, the combustible cigarettes, and the control group without nicotine. Concerning the data collected by applying the EndoPat™2000 system, we determined that the augmentation index and the reactive hyperemia index became worse after administering any type of device. More precisely, the augmentation index, representing the arterial stiffness, adjusted for a heart rate of 75 bpm (AI@75), increased significantly 15 min after JUUL™ing and remained significantly increased in the follow-up measurement 60 min after consuming the JUUL™. In comparison to the other devices, we have seen no significant differences. As lower values of the AI@75 reflect better arterial elasticity, an increase in the AI@75 can be seen as a decrease in arterial stiffness.

The reactive hyperemia index, expressing the endothelial vasodilator function, showed a significant decrease at both points of time of measurements after JUUL™ing as well as after using the other devices, which indicates acute damage of the endothelium due to JUUL™ing, heating, vaping, and smoking. These temporary hemodynamic alterations as well as changes in arterial stiffness and endothelial vasodilator function can be explained by different mechanisms. Mainly, it is triggered by nicotine. Based on our findings that the same post-vaping changes occurred in our control group (e-cigarette without nicotine) as in the others, which can also be seen in the further statistical analyses, we are led to the conclusion that there must be further triggers besides the nicotine. The potential impact of local and circulating catecholamines has already been mentioned in previous studies on e-cigarettes. Therefore, local and circulating catecholamines also reflect, among other factors, the activity and stimulation of sympathetic ganglia. Consequently, sympathetic neuronal discharge-impaired nitric oxide production increases in the central nervous system, which increases arterial stiffness [52,53,54]. By taking blood samples, we observed changes in hemogram, consisting of a rise of leucocytes, immature granulocytes, and neutrophil granulocytes 60 min after the use of any of the four devices. This can be interpreted as a part of the inflammatory response of the endothelium caused by the use of a JUUL™ device, a heated tobacco product, an e-cigarette, as well as a cigarette, as postulated by Lee et al. in his model of the pathogenesis of endothelial dysfunction triggered by e-cigarette vapor [35]. Further studies have shown that cigarette smoking leads to a leukocytosis that is reversible after smoking cessation [55,56,57]. Most of these studies have observed the increase in leucocytes as an effect of smoking in an interval of weeks after smoking, whereas our study registered the leukocytosis already in an interval of an hour after smoking, heating, vaping, or JUUL™ing. We interpret these early changes in hemogram as an acute reaction of inflammation caused by the administration of the device. Supplementary investigations considering chemokines and cytokines should be followed up. Related with the use of e-cigarettes and heated tobacco products, manifestations of acute eosinophilic pneumonia accompanied by a leukocytosis have been registered [58].

To verify this inflammation reaction, we wanted to detect a possible reaction of stress by measuring the concentration of plasma cortisol. In our study, this significantly decreased 12 min after JUUL™ing as well as two hours after the use of any of the other devices, as already shown for the conventional cigarette by Mendelsohn et al. [42]. The decrease in plasma cortisol can be interpreted as a depletion of stress hormones, representing a situation of stress for the participant after the use of the respective device.

In contrast to the current literature, the serial measurements of CO in the different devices showed a significant increase over time. In fact, no increase would have been expected especially for the e-cigarette and the JUUL™ device in the absence of combustion [59].

A conclusive explanation as to why our results differ from previous studies must be left open. Due to the low standard deviation, possible measurement errors are rather unlikely. The measurement accuracy appears to remain a possible factor, although the Micro+Smokerlyzer™ is a validated device.

Our results showed that a small part of the increase in arterial stiffness remained unexplained by the device, blood pressure, heart rate, or sex. This unexplained change might be due to the effects of the JUUL™, the HTP, the e-cigarette, and the conventional cigarette on the endothelium. Therefore, a combination of nicotine and harmful or potentially harmful compounds in smoke and vapor is one of the most likely explanations for this observation. Similar results were shown in an animal model by Nabavizadeh et al., who compared combustible cigarettes with heated tobacco products [60]. In addition, population-based studies on cardiovascular diseases found an association between e-cigarette use and increased risk of myocardial infarction [33]. However, the influence of former or current smoking remains unclear. The pathogenesis of coronary heart disease and other cardiovascular events includes stiffening of the arteries [61,62], which is an effect that could be shown here for the first time after JUUL™ing.

In addition to central hemodynamics, arterial stiffness seems to be even more important than peripheral blood pressures as a surrogate parameter for cardiovascular events [37,63,64]. It should be noted that cigarette smoke not only affects the cardiovascular system but also has other negative effects on health, and e-cigarette aerosols contain much lower amounts of harmful and potentially harmful substances than cigarette smoke [20,32,65]. Thus, a complete switch from tobacco cigarettes to e-cigarettes is widely acknowledged to reduce the exposure to many carcinogens and other toxicologically relevant compounds [21,66].

To further examine JUUL™ing as a potential risk factor for cardiovascular diseases, further trials are needed that focus on the chronic effects of the JUUL™ device and compare them with the long-term effects of conventional cigarettes and other e-cigarettes with nicotine-containing or nicotine-free liquids.

Regarding the potential sources of error of this study, it must be mentioned that the intensity of JUUL™ing, specifically the intensity of the puff, could not be standardized. To avoid large differences in vaping behavior, the frequency, number, and duration of puffs were defined. Furthermore, blood samples were carried out during the study, which could have influenced the individual stress levels of the participants and may have appeared as biases in hemodynamic as well as plasma cortisol. In addition, another weakness of the study remains the relatively small number of participants, so this should be increased in further studies. 

## 5. Conclusions

To conclude, this direct comparison of a JUUL™ device with a heated tobacco product, a conventional cigarette, and a nicotine-free e-cigarette reveals a comparable effect of each device on peripheral hemodynamics, endothelial vasodilator function, and arterial stiffness. The acute changes in arterial stiffness and endothelial vasodilator function that occur after administering the evaluated devices may lead to an increased risk for cardiovascular diseases as a consequence of a long-term consumption. However, a long-term follow-up evaluation to detect the chronic effects of JUUL™ing should be conducted.

## Figures and Tables

**Figure 1 medicines-09-00028-f001:**
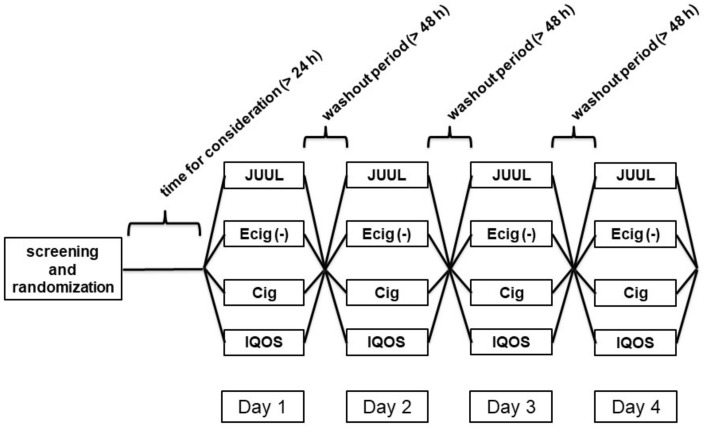
Flowchart of the study design. The minimum period to complete the study for one individual subject amounted 12 days including the screening day, the day for consideration, as well as the hours of each washout period.

**Figure 2 medicines-09-00028-f002:**
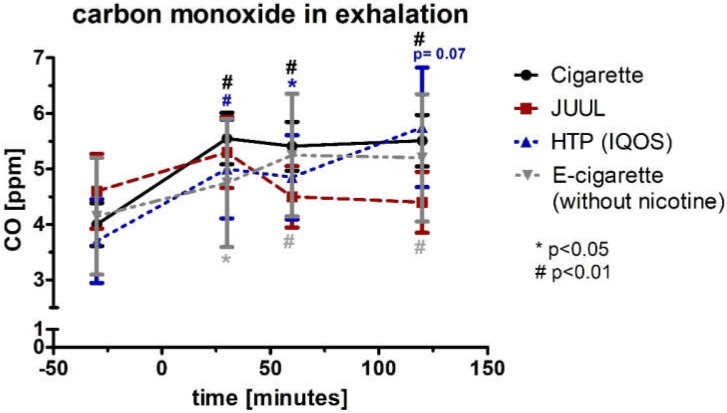
Concentration of carbon monoxide in exhaled air before and after consuming the cigarette, JUUL^TM^, heated tobacco product, and e-cigarette without nicotine.

**Figure 3 medicines-09-00028-f003:**
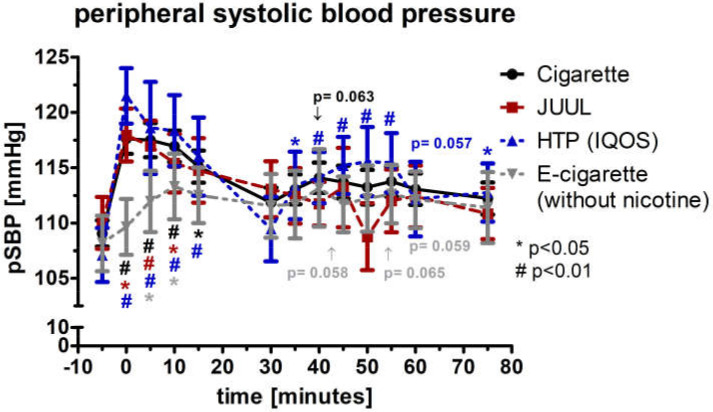
Peripheral systolic blood pressure in a frequency of 5 min until 75 min after consuming the cigarette, JUUL^TM^, heated tobacco product, and e-cigarette without nicotine. There was a significant increase in the peripheral systolic blood pressure right after the consumption of the cigarette, JUUL™, and heated tobacco product.

**Figure 4 medicines-09-00028-f004:**
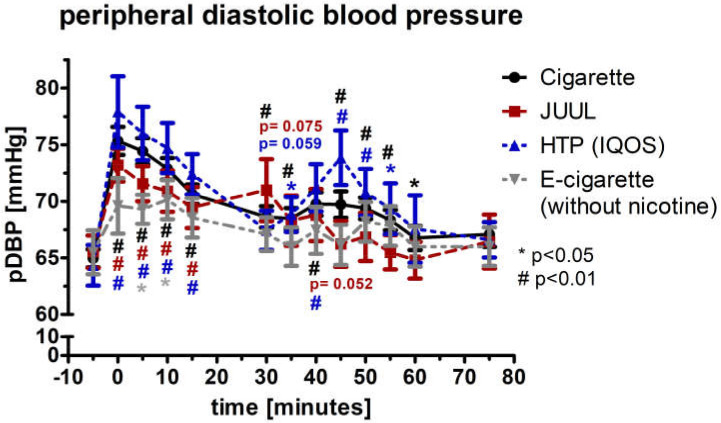
Peripheral diastolic blood pressure in a frequency of 5 min until 75 min after consuming the cigarette, JUUL^TM^, heated tobacco product, and e-cigarette without nicotine. There was a significant increase in the peripheral diastolic blood pressure right after the consumption of the cigarette, JUUL™, and heated tobacco product.

**Figure 5 medicines-09-00028-f005:**
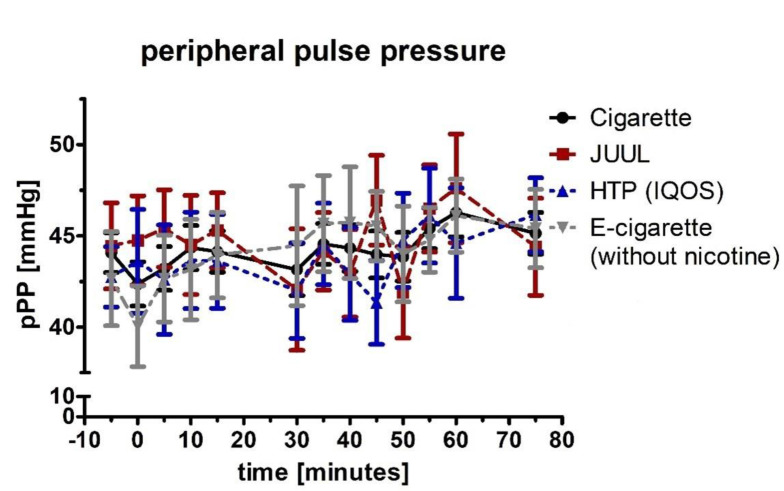
Peripheral pulse pressure in a frequency of 5 min until 75 min after consuming the cigarette, JUUL^TM^, heated tobacco product and e-cigarette without nicotine. There were no changes reaching significance after the use of any of the devices.

**Figure 6 medicines-09-00028-f006:**
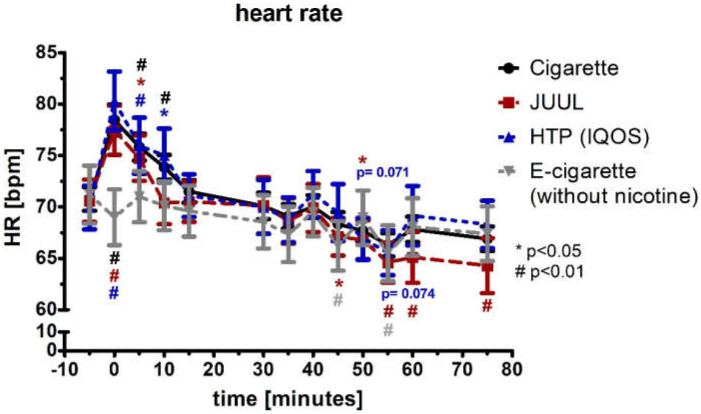
Heart rate in a frequency of 5 min until 75 min after consuming the cigarette, JUUL^TM^, heated tobacco product, and e-cigarette without nicotine. There was a significant increase in the heart rate right after the consumption of the cigarette, JUUL™, and heated tobacco product.

**Figure 7 medicines-09-00028-f007:**
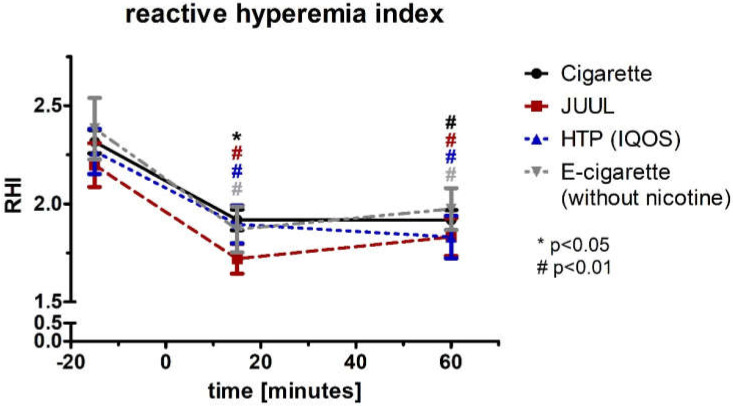
Reactive hyperemia index before and 15 as well as 60 min after consuming the cigarette, JUUL^TM^, heated tobacco product, and e-cigarette without nicotine.

**Figure 8 medicines-09-00028-f008:**
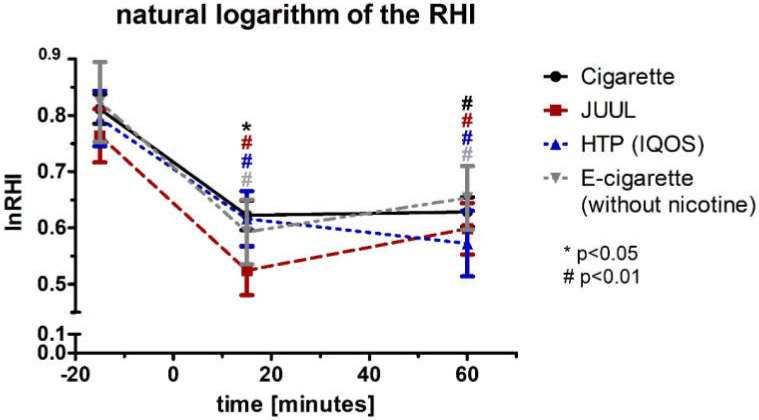
Natural logarithm of the RHI before and 15 as well as 60 min after consuming the cigarette, JUUL^TM^, heated tobacco product, and e-cigarette without nicotine.

**Figure 9 medicines-09-00028-f009:**
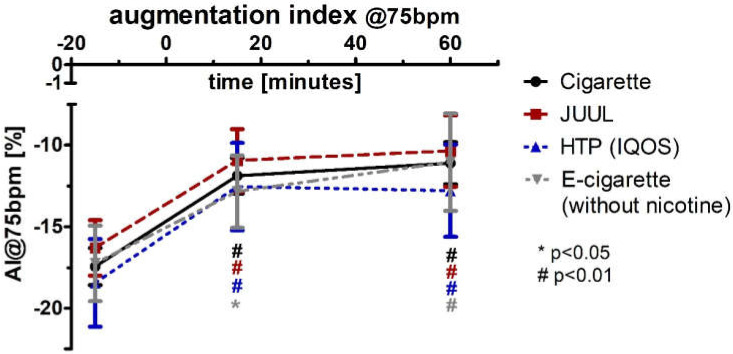
Augmentation index adjusted at a heart rate of 75 bpm before and 15 as well as 60 min after consuming the cigarette, JUUL^TM^, heated tobacco product, and e-cigarette without nicotine.

**Figure 10 medicines-09-00028-f010:**
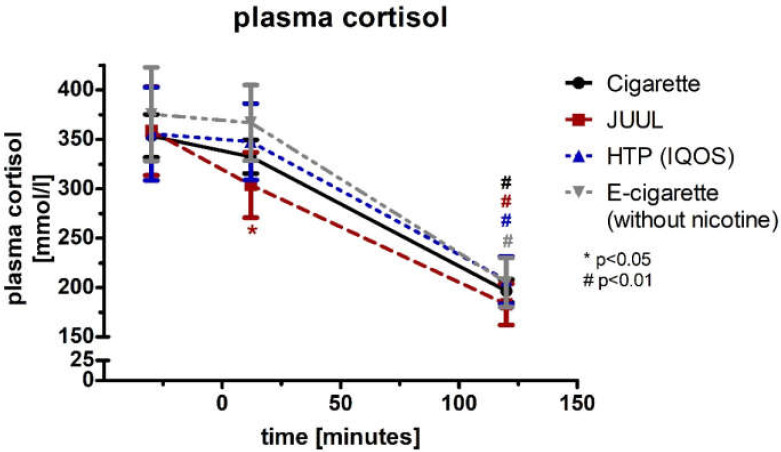
Plasma cortisol concentration analyzed before as well as 12 and 120 min after consuming the cigarette, JUUL^TM^, heated tobacco product, and e-cigarette without nicotine.

**Figure 11 medicines-09-00028-f011:**
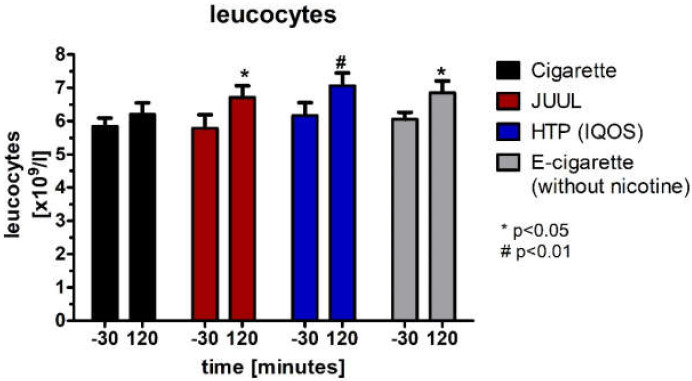
Leucocytes analyzed before, as well as 120 min after consuming the cigarette, JUUL^TM^, heated tobacco product, and e-cigarette without nicotine.

**Figure 12 medicines-09-00028-f012:**
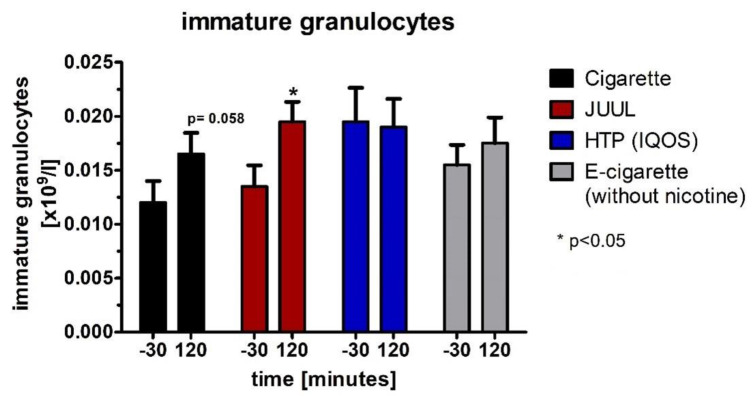
Immature granulocytes analyzed before as well as 120 min after consuming the cigarette, JUUL^TM^, heated tobacco product, and e-cigarette without nicotine.

**Figure 13 medicines-09-00028-f013:**
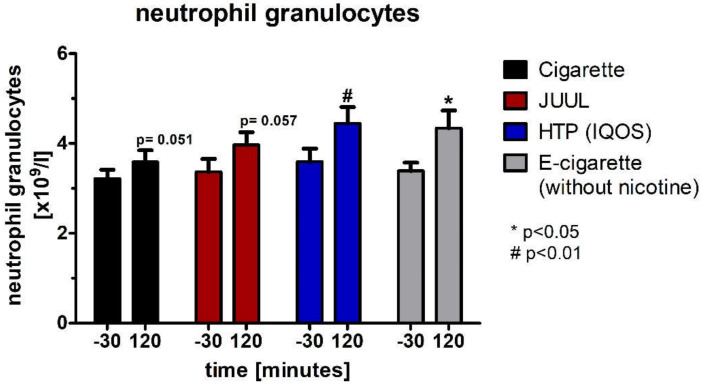
Neutrophil granulocytes analyzed before as well as 120 min after consuming the cigarette, JUUL^TM^, heated tobacco product, and e-cigarette without nicotine.

**Table 1 medicines-09-00028-t001:** Baseline characteristics for all analyzed 20 participants.

Sex	All subjects (*n* = 20)	Male (*n* = 10)	Female (*n* = 10)	*p*-Value
	**Mean ± SED**	**Mean ± SED**	**Mean ± SED**	
**Age [years]**	25.2 ± 0.9	25.6 ± 0.4	24.7 ± 0.3	0.6063
**Weight [kg]**	78.2 ± 3.5	85.9 ± 1.9	68.8 ± 0.8	0.0104
**Height [m]**	1.76 ± 0.0	1.8 ± 0.0	1.7 ± 0.0	0.0040
**BMI [kg/m²]**	25.0 ± 0.8	25.9 ± 0.4	24.0 ± 0.2	0.2180
**Waist [cm]**	82.1 ± 2.0	86.1 ± 0.7	77.1 ± 0.7	0.0181
**Hip [cm]**	92.9 ± 2.0	93.5 ± 1.0	92.0 ± 0.7	0.7155
**Cigarettes per day**	2.1 ± 0.7	2.7 ± 0.1	1.2 ± 0.2	0.2832
**Fagerström-Test [points]**	0.5 ± 0.2	0.8 ± 0.1	0.1 ± 0.0	0.1180

**Table 2 medicines-09-00028-t002:** Measurements of arterial stiffness represented by the reactive hyperemia index (RHI) and its natural logarithm (LnRHI) and endothelial vasodilator function represented by the augmentation index (AI) adjusted at a heart rate of 75 beats per minute (AI@75bpm).

		Combustible Cigarette		JUUL^TM^		Heated Tobacco Product		E-Cigarette		
	**Time [min]**	**Mean ± SED**	**Paired *t*-Test**	**Mean ± SED**	**Paired *t*-Test**	**Mean ± SED**	**Paired *t*-Test**	**Mean ±** **SED**	**Paired *t*-Test**	**ANOVA**
**RHI**	**Baseline**	2.3 ± 0.06		2.2 ± 0.11		2.3 ± 0.11		2.4 ± 0.16		0.540
	**15**	1.9 ± 0.05	0.032	1.7 ± 0.08	0.001	1.9 ± 0.1	0.001	1.9 ± 0.12	0.002	0.016
	**60**	1.9 ± 0.05	0.001	10.8 ± 0.10	0.001	10.8 ± 0.11	0.004	20.0 ± 0.11	0.000	0.368
**LnRHI**	**Baseline**	0.8 ± 0.03		0.8 ± 0.048		0.8 ± 0.05		0.8 ± 0.07		0.587
	**15**	0.6 ± 0.03	0.022	0.5 ± 0.04	0.000	0.6 ± 0.05	0.002	0.6 ± 0.06	0.002	0.017
	**0**	0.6 ± 0.03	0.001	0.6 ± 0.05	0.001	0.6 ± 0.06	0.003	0.7 ± 0.06	0.000	0.363
**AI@75bpm [%]**	**Baseline**	−17.4 ± 10.1		−16.3 ± 1.7		−18.5 ± 2.7		−17.3 ± 2.31		0.928
	**15**	−11.9 ± 10.1	0.000	−110.0 ± 1.9	0.000	−120.6 ± 2.7	0.000	−12.9 ± 2.21	0.013	0.907
	**60**	−110.1 ± 1.3	0.001	−10.4 ± 2.2	0.002	−120.8 ± 20.8	0.000	−110.1 ± 2.99	0.008	0.888

**Table 3 medicines-09-00028-t003:** Plasma cortisol concentration analyzed 12 and 120 min after consuming the cigarette, JUUL^TM^, heated tobacco product, and e-cigarette without nicotine.

Plasma Cortisol [mmol/L]									
	Combustible Cigarette		JUUL^TM^		Heated Tobacco Product		E-Cigarette		
**Time [min]**	**Mean ± SED**	**paired *t*-test**	**Mean ± SED**	**paired *t*-test**	**Mean ± SED**	**paired *t*-test**	**Mean** ± **SED**	**paired *t*-test**	**ANOVA**
**Baseline**	353.5 ± 21.7		358.5 ± 44.8		355.6 ± 47.1		375.5 ± 47.4		0.873
**12**	332.5 ± 17.1	0.571	303.9 ± 33.2	0.049	347.6 ± 38.7	0.792	366.8 ± 38.2	0.786	0.527
**120**	196.6 ± 11.6	0.000	183.0 ± 20.9	0.000	207.4 ± 24.0	0.000	205.1 ± 24.9	0.000	0.865

**Table 4 medicines-09-00028-t004:** Changes within the hemogram analyzed before as well as 120 min after consuming the cigarette, JUUL^TM^, heated tobacco product, and e-cigarette without nicotine.

	Hemogram									
		Combustible Cigarette		JUUL^TM^		Heated Tobacco Product		E-Cigarette		
**Type of cell**	**Time [min]**	**Mean ± SED**	**paired *t*-test**	**Mean ± SED**	**paired *t*-test**	**Mean ± SED**	**paired *t*-test**	**Mean ± SED**	**paired *t*-test**	**ANOVA**
**Leucocytes [G/l]**	Baseline	5.8 ± 0.3		5.8 ± 0.4		6.2 ± 0.4		6.1 ± 0.2		0.816
	120	6.2 ± 0.3	0.090	6.7 ± 0.4	0.047	7.1 ± 0.4	0.003	6.9 ± 0.4	0.024	0.374
**Immature granulocytes [G/l]**	Baseline	0.012 ± 0.0		0.014 ± 0.0		0.02 ± 0.0		0.016 ± 0.0		0.119
	120	0.017 ± 0.0	0.058	0.02 ± 0.0	0.014	0.019 ± 0.0	0.847	0.018 ± 0.0	0.447	0.765
**Neutrophil granulocytes [G/l]**	Baseline	3.2 ± 0.2		3.4 ± 0.3		3.6 ± 0.3		3.4 ± 0.2		0.743
	120	3.6 ± 0.3	0.051	4.0 ± 0.3	0.057	4.4 ± 0.4	0.006	4.3 ± 0.4	0.034	0.237

## Data Availability

Not applicable.

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
