# Peer review of "JUUL™ing and Heating Lead to a Worsening of Arterial Stiffness"

_medicines, 2022, doi:10.3390/medicines9040028_

Round 1
Reviewer 1 Report
First of all, thanks for the opportunity to review this paper.
The authors present an original article about the effects of Juuling and heating on arterial stiffness.
Overall it is an interesting paper and congrats to the authors.
I have though 2 minor comments:
- The figures’ format is to small making it almost illegible
- Please add brief comments to figures and tables and most importantly when acronyms are used.
Author Response
- The figures’ format is too small making it almost illegible.
- That’s an important piece of advice. We’ve improved the resolution to 600 dpi and extended the figure’s format. In addition, we considered excluding the error bars for the figures of the systolic and diastolic blood pressure as well as the heart rate that seemed to be overloaded by the error bars due to the frequent measurement time point. Finally, we’ve decided to include the error bars because they give important information of diversity, that shouldn’t be withheld from the reader.
- Please add brief comments to figures and tables and most importantly when acronyms are used.
- Good point. We’ve added comments to each figure and table. We’ve used the long version to make the figures more comprehensible according to the acronyms.
Reviewer 2 Report
Thank you for permitting me to review this manuscript
Abstract
Please expanse which parameters were used in the assessment of arterial stiffness
Introduction :
Line 59 : Please describe in a brief sentence EVALI
Line 104 : Please rephrase starting with the primary objective of this study etc..
PLease tell how many patients participated exactly or finally to the study , as I understant those who did not perform all 4 test were deleted?
Figure 1 please define total number of days for the study , 11?
Which test is used for thos parameters which were not normalyy distributed ?
Figure 2-60 should be presented with a better resolution it is very hard to differentiate the curves
Methods
This study would have benefited from control groups of non previously smoking of individuals to estabilish some kind of normal values
How are we sure that the effect of each component would have disappeared after 48 h ? therefore the crossover design should be discussed
Author Response
- Concerning the abstract: Please expanse which parameters were used in the assessment of arterial stiffness.
– Definitely, those parameters are important to mention. Therefore, we’ve added them directly after mentioning the term “parameters of arterial stiffness”.
- Concerning the introduction:
a) Line 59 : Please describe in a brief sentence EVALI
- We’ve tried to make it as brief as possible. Nevertheless, the sentence turned out to become extensive as the four different criteria of the definition of EVLAI needed to be mentioned. (line 62 – 67)
b) Line 104 : Please rephrase starting with the primary objective of this study etc..
- That is a great suggestion. By rephrasing the sentence, the objectives of our study are now more apparent and hopefully more concisely. (line 110 and following) - Please tell how many patients participated exactly or finally to the study , as I understant those who did not perform all 4 test were deleted?
- In total, 20 subjects participated completely. Therefore, there were 80 measurements performed and included in the statistical analysis. You’ve understood it right, that there were three participants besides the 20 subjects, that were regarded as dropouts because they didn’t complete all four test conditions. - Figure 1 please define total number of days for the study , 11?
- For a single participant, the minimum period to complete the study amounted to 12 days including the screening day, the day for consideration, as well as the hours of each washout period.
- Which test is used for those parameters which were not normally distributed?
- The Wilcoxon-test was used for those parameters. We’ve added this important information to those parameters.
- Figure 2-60 should be presented with a better resolution it is very hard to differentiate the curves. - That’s an important piece of advice. We’ve improved the resolution to 600 dpi and extended the figure’s format. In addition, we considered excluding the error bars for the figures of the systolic and diastolic blood pressure as well as the heart rate that seemed to be overloaded by the error bars due to the frequent measurement time point. Finally, we’ve decided to include the error bars because they give important information of diversity, that shouldn’t be withheld from the reader.
- Concerning the methods: This study would have benefited from control groups of non previously smoking of individuals to establish some kind of normal values.
-Thank you for this consideration. The study would indeed gain more importance if we could have conducted the measurements with non-smokers. Unfortunately, the local ethics committee wouldn't have approved it if we would have asked non-smokers to participate. Even a single use of nicotine has addictive potential, so participation in the study could have given individual subjects a taste for smoking. In addition, likely, the non-smokers would not have inhaled the smoke and vapor as intensely as necessary for the study. However, it is important to mention that all participants were young and healthy, and most of them were only occasional smokers, so it cannot be assumed that there was preexisting vascular damage due to smoking.
- How are we sure that the effect of each component would have disappeared after 48 h? therefore the crossover design should be discussed.
- Important remark. Concerning the washout period of nicotine, there is proof in several studies, that most people can expect nicotine to be fully eliminated from their system within 24 hours. Since nicotine has a half-life of approximately 2 hours, it theoretically should be cleared from your body within 11 hours. Concerning the other components, you’re right, we cannot be entirely sure, that the effect of each component terminated after 48 hours because literature about the washout period of these components is still limited yet.
Round 2
Reviewer 2 Report
The authors has significantly improved the manuscript and responded to my queries